# LITESEC-T3SS - Light-controlled protein delivery into eukaryotic cells with high spatial and temporal resolution

Florian Lindner[1], Bailey Milne-Davies[1], Katja Langenfeld [1], Thorsten Stiewe [2] & Andreas Diepold [1✉]

Many bacteria employ a type III secretion system (T3SS) injectisome to translocate proteins into eukaryotic host cells. Although the T3SS can efficiently export heterologous cargo proteins, a lack of target cell specificity currently limits its application in biotechnology and healthcare. In this study, we exploit the dynamic nature of the T3SS to govern its activity. Using optogenetic interaction switches to control the availability of the dynamic cytosolic T3SS component SctQ, T3SS-dependent effector secretion can be regulated by light. The resulting system, LITESEC-T3SS (Light-induced translocation of effectors through sequestration of endogenous components of the T3SS), allows rapid, specific, and reversible activation or deactivation of the T3SS upon illumination. We demonstrate the light-regulated translocation of heterologous reporter proteins, and induction of apoptosis in cultured eukaryotic cells. LITESEC-T3SS constitutes a new method to control protein secretion and translocation into eukaryotic host cells with unparalleled spatial and temporal resolution.

[1] Department of Ecophysiology, Max Planck Institute for Terrestrial Microbiology, Karl-von-Frisch-Str. 10, 35043 Marburg, Germany. [2] Institute of Molecular Oncology, Member of the German Center for Lung Research (DZL), Philipps-University, Hans-Meerwein-Str. 3, 35043 Marburg, Germany. ✉email: andreas.diepold@mpi-marburg.mpg.de

The injectisome is a bacterial nanomachine capable of translocating proteins into eukaryotic host cells in a one-step export mechanism[1,2]. The core components of the injectisome, or type III secretion system (T3SS) are shared with the bacterial flagellum[3,4]. In this manuscript, T3SS refers to the virulence-associated T3SS. The common Sct nomenclature[5] is used for T3SS components; see ref. [6] for species-specific names. The injectisome consists of (i) an extracellular needle formed by helical polymerization of a small protein and terminated by a pentameric tip structure, (ii) a series of membrane rings that span both bacterial membranes and embed (iii) the export apparatus, formed by five highly conserved hydrophobic proteins thought to gate the export process, and (iv) a set of essential cytosolic components, which cooperate in substrate selection and export (Fig. 1a).

The injectisome is an essential virulence factor for many pathogenic Gram-negative bacteria, including *Salmonella*, *Shigella*, pathogenic *Escherichia coli*, and *Yersinia*[7]. It is usually assembled upon entry into a host organism, but remains inactive until contact to a host cell has been established. At this point, the injectisome exports two translocator proteins that form a pore in the host membrane, and a pool of so-called T3SS effector proteins that are translocated into the host cell.

The Gram-negative enterobacterium *Yersinia enterocolitica* uses the T3SS to translocate six Yop (*Yersinia* outer protein) effector proteins into phagocytes, which prevent phagocytosis and block proinflammatory signaling[8]. In this study, we use the *Y. enterocolitica* strain IML421asd (ΔHOPEMTasd)[9], where these six virulence effectors have been deleted, and which is additionally auxotrophic for the cell wall component diaminopimelic acid. The strain is therefore nonpathogenic, but possesses a functional T3SS. Secretion of effector proteins can be triggered in vivo by host cell contact or in vitro by low $Ca^{2+}$ levels in the medium[10].

Being a machinery that evolved to efficiently translocate proteins into eukaryotic cells, the T3SS has been successfully used to deliver protein cargo into a wide variety of eukaryotic target cells for different purposes such as vaccination, immunotherapy, and gene editing (reviewed in ref. [11]). Export through the T3SS is fast

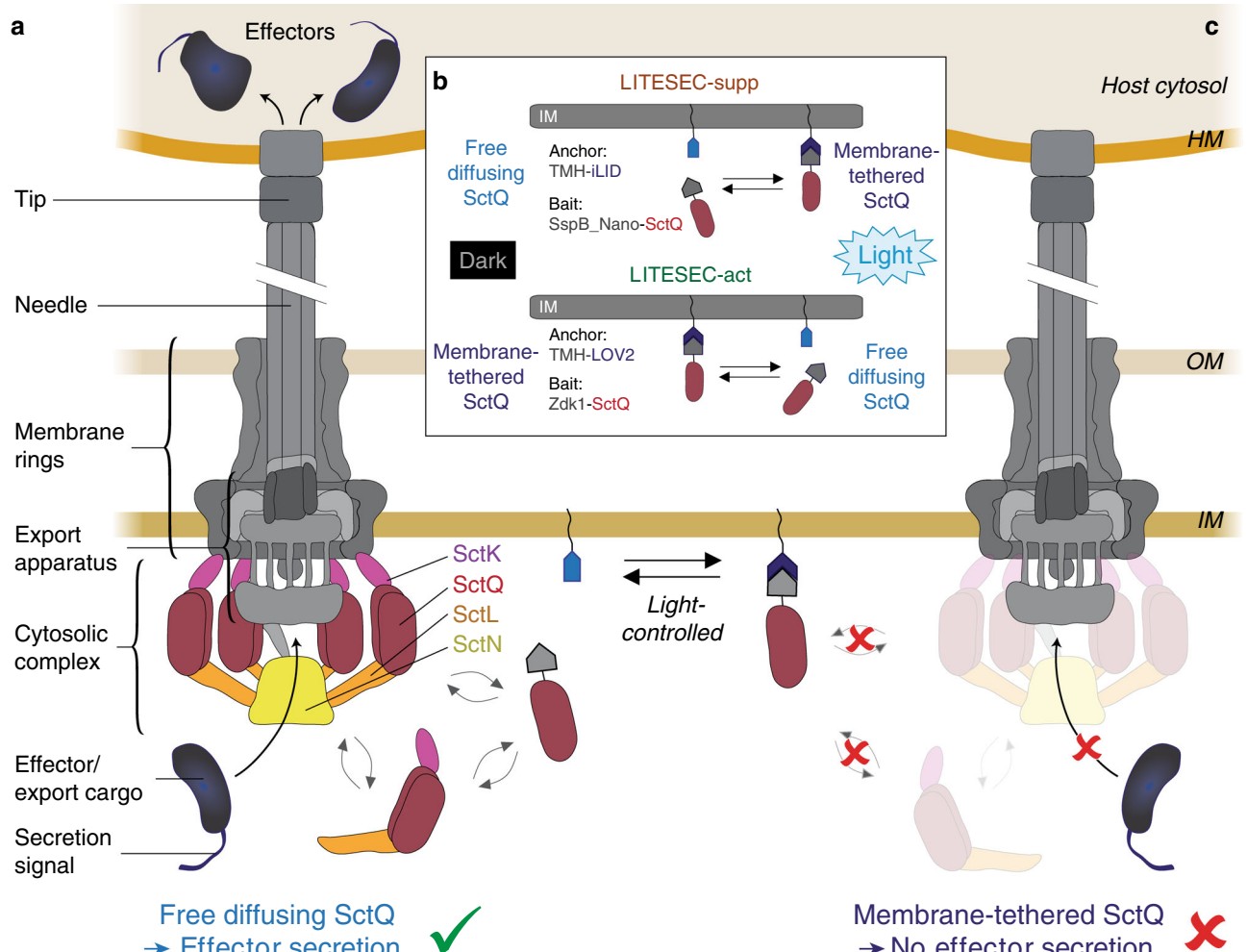

**Fig. 1 Working principle of the LITESEC systems—light-controlled activation and deactivation of protein translocation by the type III secretion system. a** Schematic representation of the active T3SS injectisome (modified from ref. [6]). Left side, main substructures; right side, dynamic cytosolic T3SS components. Effector translocation by the T3SS is licensed by the functional interaction of the unbound bait-SctQ fusion with the T3SS. **b** Different states of the bait and anchor proteins in dark and light conditions. In the LITESEC-supp system (top), the bait protein, a fusion of the smaller interaction switch domain SspB_Nano and the essential T3SS component SctQ, is tethered to the inner membrane (IM) by a membrane anchor, a fusion of a transmembrane helix (TMH) and the larger interaction switch domain, iLID, in the light, and gets released in the dark. Conversely, in the LITESEC-act system (bottom), the bait protein, a fusion of the smaller interaction switch domain, Zdk1, and the essential T3SS component SctQ, is tethered to the membrane anchor, a TMH fusion of the larger interaction switch domain, LOV2, in the dark, and gets released by illumination. **c** Sequestration of the bait-SctQ fusion protein to the membrane prevents effector secretion. HM host membrane, OM bacterial outer membrane, IM bacterial inner membrane.

and efficient: More than $10^6$ effectors can be translocated into a single host cell at rates of several hundred effectors per second for one injectisome[12–15]. Short N-terminal secretion signals mark cargo proteins for delivery by the T3SS[16,17]. The size and structure of the cargo proteins can influence translocation rates, and very large or stably folded proteins (such as GFP or dihydrofolate reductase) are exported at a lower rate. However, most cargoes, including large proteins with molecular weights above 60 kDa, can be exported by the T3SS[15,18,19]. Protein translocation into host cells can be titrated by adjusting the expression level and multiplicity of infection (ratio of bacteria and host cells). Within the host, the T3SS secretion signal can be removed by site-specific proteases or cleavage at the C-terminus of a ubiquitin domain by the native host cell machinery, and subcellular localization can be influenced using nanobodies cotranslocated by the T3SS[15,20]. Taken together, these properties make the T3SS an efficient and versatile tool for protein delivery into eukaryotic cells[11,15].

T3SS inject effector proteins into any eukaryotic host cell as soon as they are in contact. Lack of target specificity is therefore a main obstacle in the further development and application of T3SS-based protein delivery systems[21,22].

Four soluble cytosolic components of the T3SS (SctK, L, Q, N) form an interdependent complex at the proximal interface of the injectisome[23–30] (Fig. 1a). As these proteins interact with effectors and their chaperones with a graded affinity matching the export order of the effectors, they were termed sorting platform[31]. Our group recently discovered that the sorting platform proteins of the *Y. enterocolitica* T3SS constantly exchange between the injectisome and a cytosolic pool (Fig. 1a), and that this exchange is linked to protein secretion by the T3SS[26,32]. We rationalized that the constant shuttling of these essential T3SS components might allow to control T3SS activity through reversible sequestration of one of the cytosolic proteins, thereby establishing a completely new way of regulating the T3SS.

Optogenetics combines optical and genetic methods to precisely and reversibly control gain or loss of protein function in living cells or tissues. It allows fast (within milliseconds) and specific (to single proteins) control of defined events in biological systems[33], giving optogenetic approaches an advantage over knockdown, overexpression, or mutant strain analysis, which often display slower activation and a broader effect[34]. Optogenetic protein interaction switches use light-induced conformational changes of specific proteins, often light-oxygen-voltage (LOV) domain proteins, to control protein–protein interactions by light[35,36]. They usually consist of homo- or hetero-dimers whose affinities are strongly altered upon irradiation by light of a certain wavelength. Mutations of specific amino acids in the optogenetic interaction domains can modulate binding affinities and the corresponding dissociation or return rates from a few seconds to several minutes[36,37].

Optogenetic interaction switches were established and have mainly been studied in eukaryotic cells[38]. In this work, we therefore tested the applicability of two different optogenetic interaction switches in bacteria: (i) The LOVTRAP system (LOV), which consists of the two interacting proteins LOV2 (a photo sensor domain from *Avena sativa* phototropin 1) and Zdk1 (Z subunit of the protein A), that bind to each other in the dark. Upon irradiation with blue light, LOV2 undergoes a conformational change and Zdk1 is released[36]. (ii) The iLID system, which employs the interaction of iLID, derived from a LOV2 domain from *Avena sativa* phototropin 1, with a smaller binding partner, SspB_Nano. The iLID system has a low binding affinity in the dark and a high affinity upon irradiation with blue light[35,37]. LOV and iLID systems therefore react to light in opposite directions, which allows to specifically release a bait protein (and, subsequently, to activate processes

that require its presence) in the dark or upon illumination, respectively.

To establish the use of optogenetic interaction switches in bacteria, we first assessed the effect of illumination on the different switches by light microscopy, using fluorescently labeled bait proteins. Next, we applied the switches to control the availability of the essential cytosolic T3SS component SctQ and, in consequence, secretion of cargo proteins through the T3SS, by light. We optimized the systems by defining suitable versions of the switches and adjusting the expression ratio of anchor and bait proteins. As proof of concept, we show the light-dependent translocation of heterologous cargo proteins into eukaryotic host cells. The successful development of the LITESEC system presents a blueprint for the application of optogenetic interaction switches in prokaryotes, and opens widespread opportunities for using the T3SS as a specific and precisely controllable tool to deliver proteins of interest into eukaryotic cells.

## Results

**Controlling the activity of the T3SS with light**. To establish a method to control protein translocation by the T3SS, we took advantage of our recent finding that some essential cytosolic T3SS components constantly exchange between the cytosol and the injectisome[26,32]. We combined one of these components, SctQ, with one partner domain of an optogenetic interaction switch, and targeted the other partner domain to the bacterial inner membrane (IM) by adding an N-terminal transmembrane helix. We reasoned that this might allow to control SctQ availability in the cytosol, and therefore T3SS-based protein export and translocation into host cells, by light. To be able to control T3SS activity in both directions, we developed two complementary systems:

(a) LITESEC-supp, a system that confers suppression of T3SS-dependent protein translocation by blue light illumination.
(b) LITESEC-act, a system that confers activation of T3SS-dependent protein translocation by blue light illumination.

Both systems rely on two interaction partners which we have engineered:

(i) A membrane-bound anchor protein, which is a fusion between the N-terminal transmembrane helix (TMH) of a well-characterized transmembrane protein, *Escherichia coli* TatA, extended by two amino acids (Val-Leu) for more stable insertion in the IM, a Flag peptide for detection and spacing, and the larger domain of the respective optogenetic interaction switches, iLID (for LITESEC-supp) or LOV2 (for LITESEC-act). The resulting fusion proteins, TMH-iLID / TMH-LOV2, are expressed from a plasmid.
(ii) A bait protein, which consists of a fusion between the essential cytosolic T3SS component SctQ and the smaller domain of the interaction switches, SspB_Nano (LITESEC-supp)/Zdk1 (LITESEC-act). Based on the estimated cellular SctQ concentration of 1−2 μM (approximately 1000 molecules per bacterium)[26], we chose SspB_nano, which has a dissociation constant of 132 nM in the light and 4.7 μM in the dark, over SspB_micro and SspB_milli, which have roughly 10 and 1000 times higher dissociation constants[37]. The resulting fusion proteins, SspB_Nano-SctQ/Zdk1-SctQ, replace the wild-type SctQ protein on the *Y. enterocolitica* virulence plasmid by allelic exchange of the genes[39].

Coexpression of both interaction partners provides the basis for light-controlled protein translocation by the T3SS (Fig. 1). For the iLID-based LITESEC-supp system, the bait protein is tethered

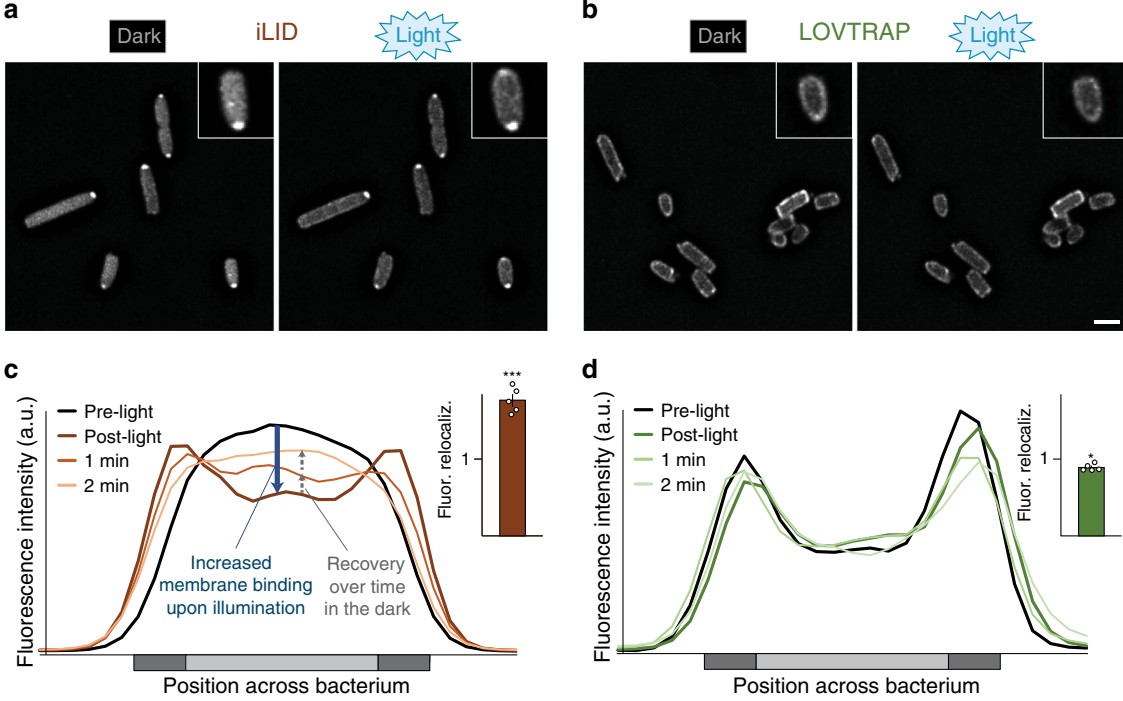

**Fig. 2 Activation and recovery kinetics of optogenetic sequestration systems. a, b** Fluorescence micrographs of mCherry-labeled bait proteins in the iLID-based (**a**) and LOV-based (**b**) sequestration systems, before (left) and directly after (right) illumination with blue light. Size bar, 2 μm; insets 2× enlarged. **c, d** Representative fluorescence signal quantification across bacteria over time in the iLID-based (**c**) and LOV-based (**d**) sequestration systems; dark gray: membrane, light gray: cytosol. Insets: Fluorescence relocalization factor (fluor. reloc. $= R_{\text{post-light}}/R_{\text{pre-light}}$, where $R$ represents the ratio of fluorescence intensities at the membrane and in the cytosol, before and after illumination, respectively), based on 121−131 line scans across five cells per strain and time point. Single relocalization factor values ($n = 5$) indicated by circles. Bars show mean values; error bars represent the standard deviation, *$p = 0.0002$; ***$p = 6.2 \times 10^{-6}$ against no relocalization in a two-tailed homoscedastic $t$ test. Source data for panels (**c**) and (**d**) are provided as a Source Data File.

to the membrane anchor in the light, and SctQ is therefore not available to interact with the T3SS (Fig. 1b). As SctQ is essential for the function of the T3SS, protein secretion by the T3SS is prevented (Fig. 1c). In the dark, the bait protein is not bound to the membrane, and can therefore functionally interact with the T3SS, allowing protein secretion by the T3SS (Fig. 1a). Conversely, in the LOV-based LITESEC-act system, the bait protein is released from the membrane upon irradiation with blue light, licensing protein secretion by the T3SS (Fig. 1).

**Optogenetic sequestration systems in *Y. enterocolitica*.** To assess the function and efficiency of the used optogenetic interaction switches as sequestration systems in prokaryotes, and to monitor their dynamics, we visualized the components of iLID- and LOV-based sequestration systems[35,36] in live *Y. enterocolitica* by time-lapse fluorescence microscopy. We coexpressed the anchor protein with a version of the corresponding bait protein where SctQ was replaced by mCherry to allow for a characterization of the switch by fluorescence microscopy. Initially, we confirmed that mCherry fused to the membrane anchor showed a strict membrane localization (Supplementary Fig. 1), indicating a stable fusion and a functional TMH motif. Next, the localization of mCherry-bait fusions was determined by fluorescence microscopy in live *Y. enterocolitica* expressing the corresponding unlabeled anchor proteins. Bacteria were grown in the dark and the distribution of the bait proteins was monitored before and after a short pulse of blue light (Fig. 2a, b). To quantify the change of the normalized fluorescence signal across the bacterial cells, line scans were performed (Fig. 2c, d). For the iLID system, the fluorescence signal of the bait-mCherry was cytosolic in the

preactivated state. After activation of the interaction switch with blue light, the fluorescence signal partly shifted to the membrane (Fig. 2a) and returned to the cytosol within the next minutes (Fig. 2c). In contrast, for the LOV-based sequestration system, the fluorescence signal of the bait-mCherry was mainly membrane localized in the preactivated state. Activation with blue light led to only a minor relocalization of the signal from the membranes to the cytosol (Fig. 2b, d), suggesting that the majority of bait protein remained bound to the anchor even after illumination.

**Development and characterization of LITESEC strains.** For the development of the LITESEC strains, we replaced SctQ with the bait fusion proteins Zdk1-SctQ or SspB_Nano-SctQ at its native genetic location via allelic exchange. We confirmed the stability of the fusion proteins in the LITESEC strains by Western blot (Supplementary Fig. 2). Protein secretion in wild-type *Y. enterocolitica* was not influenced by the used illumination (Supplementary Fig. 3A), and the blue light had no influence on the growth of *Y. enterocolitica* (Supplementary Fig. 3B).

**Inhibition of protein secretion by light in LITESEC-supp.** Can we use LITESEC to control T3SS secretion by light? We first tested the LITESEC-supp1 system, designed to suppress T3SS protein secretion upon illumination, in an in vitro protein secretion assay under conditions that usually lead to effector secretion (37 °C, presence of 5 mM ethylene glycol-bis(β-aminoethyl ether)-N,N,N′,N′-tetraacetic acid (EGTA) in the medium)[10]. The control strain lacking the membrane anchor secreted effectors irrespective of the illumination (Fig. 3, lanes 4, 5),

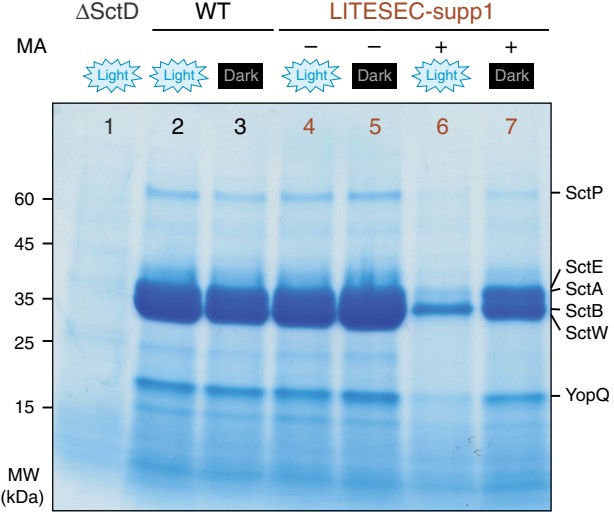

**Fig. 3 Secretion of effector proteins by the type III secretion system can be controlled by light.** In vitro secretion assay showing light-dependent export of native T3SS substrates (indicated on the right) in the LITESEC-supp1 strain. Proteins secreted by $3 \times 10^8$ bacteria during a 180-min incubation period were precipitated and analyzed by sodium dodecyl sulfate–polyacrylamide gel electrophoresis (SDS-PAGE). A strain lacking the membrane anchor (MA), the wild-type strain ΔHOPEMTasd and the T3SS-negative strain ΔSctD were used as controls. MW molecular weight in kDa.

confirming the functionality of the used SctQ fusion protein. Strikingly, the LITESEC-supp1 system showed a high level of secretion when grown in the dark, but strongly reduced secretion when grown under blue light (Fig. 3, lanes 6, 7). To quantify the difference of secretion under light and dark conditions, we define the light/dark secretion ratio (L/D ratio) as the ratio of secretion efficiency under light and dark conditions. For the LITESEC-supp1 system, the L/D ratio was 0.28, with normalized secretion efficiencies of $23.5 \pm 8.1\%$ and $85.1 \pm 5.1\%$ in light and dark conditions, respectively.

**Improvement of LITESEC-act functionality.** We next tested the LITESEC-act1 system, designed for induction of secretion by blue light illumination, and detected only a very weak activation of protein export under light conditions (Fig. 4, lanes 1, 2). Based on the fact that secretion was wild-type-like in the absence of the membrane anchor (Fig. 4, lane 7), and the results of the earlier sequestration experiment (Fig. 2b, d), we concluded that bait and anchor interact too strongly in the LITESEC-act1 system. Therefore, we constructed and tested additional versions of the system, using the mutated anchor version V416L, which displays a weaker affinity to the bait[36]. We hypothesized that a lower anchor/bait expression ratio could additionally lead to more efficient release of the bait and activation of T3SS secretion upon illumination, and expressed the V416L version of the anchor both from the medium-high expression pBAD vector used previously (LITESEC-act2), and a constitutive low-expression vector, pACYC184 (LITESEC-act3). We confirmed that the anchor proteins expressed from the pBAD plasmids show a higher expression level than the anchor proteins expressed from the pACYC184 plasmid (Supplementary Fig. 4). The response of the resulting LITESEC systems (Fig. 4c) to light was tested in an in vitro secretion assay. LITESEC-act2 showed significant induction of protein secretion in the light, compared to dark conditions (L/D ratio 2.16, Fig. 4, lanes 3, 4). Even more markedly, LITESEC-act3 allowed an almost complete activation of

secretion upon illumination (L/D ratio 4.18, Fig. 4, lanes 5, 6). Both new strains retained the low level of export in the dark. We also expressed the anchor for the LITESEC-supp system from pACYC184. The resulting LITESEC-supp2 system showed efficient secretion in the dark and strong suppression of secretion upon illumination (L/D ratio 0.26), comparable with the LITESEC-supp1 system (Fig. 4, lanes 8−11). Based on their strong reaction to light, low secretion under "off" conditions, and compatibility with the inducible pBAD expression vector, we used the LITESEC-act3 and -supp2 versions for the following experiments, unless noted otherwise. To additionally characterize the influence of the light intensity, we tested the secretion activity under ambient light conditions, where LITESEC-act3 showed intermediate secretion efficiency, whereas LITESEC-supp2 displayed almost full secretion levels (Supplementary Fig. 5).

**Light-dependent T3SS activation depends on anchor/bait ratio.** To more thoroughly explore the connection between the anchor/bait expression ratio and the responsiveness of the T3SS to illumination, we compared the secretion levels under light and dark conditions for different expression levels of the anchor in the LITESEC-act2 system. The results show that indeed, the light responsiveness of the system (the difference between secretion levels under light and dark conditions) was optimal for intermediate anchor expression levels (Fig. 5), Supplementary Fig. 6.

**Light-controlled export of heterologous T3SS substrates.** The T3SS-dependent export of heterologous cargo has been shown and applied for many purposes in earlier studies[11,15,21]. To confirm that we can control the export of heterologous proteins in the LITESEC strains, we combined the LITESEC-act3 and -supp2 systems with a plasmid expressing a heterologous cargo protein, the luciferase NanoLuc, fused to a short N-terminal secretion signal, $\text{YopE}_{1-53}$[17,40,41], and a C-terminal FLAG tag for detection. The cargo protein was exclusively exported in light conditions by the LITESEC-act3 strain, and exclusively in the dark by the LITESEC-supp2 strain, whereas export was light-independent in a wild-type strain (Fig. 6).

**Kinetics of light-induced T3SS activation and inactivation.** How efficiently can the LITESEC system be inactivated, and what are the kinetics of light-induced T3SS activation and deactivation? Protein secretion for the LITESEC-act3 and -supp2 strains was analyzed for bacteria incubated consecutively for 60 min under inactivating conditions (dark for LITESEC-act3, light for LITESEC-supp2), 60 min under activating conditions, and another 60 min under inactivating conditions. After each incubation period, the culture medium was replaced, and a sample was tested for secretion. Secretion of the heterologous export substrate $\text{YopE}_{1-53}$-NanoLuc-FLAG in LITESEC-act3 was specifically induced in light conditions, and efficiently suppressed in the dark, whereas LITESEC-supp2 displayed the opposite behavior (Fig. 7a). Similar results were obtained for native secretion substrates (Supplementary Fig. 7). The WT strains continuously secreted proteins irrespective of the illumination. These results show that the activity of the LITESEC systems can be efficiently toggled. Besides the Western blot, we used a sensitive bioluminescence-based luciferase assay[42] to quantify the export efficiency of the reporter protein $\text{YopE}_{1-53}$-NanoLuc-FLAG (Fig. 7b). We also used this assay to more precisely determine the activation and deactivation kinetics of the LITESEC system under changing illumination. In the LITESEC-supp2 strain, secretion of the heterologous substrate dropped to background levels within 4−8 min after the start of blue light illumination, and recovered within the first 4 min after shifting the bacteria to dark conditions again. The

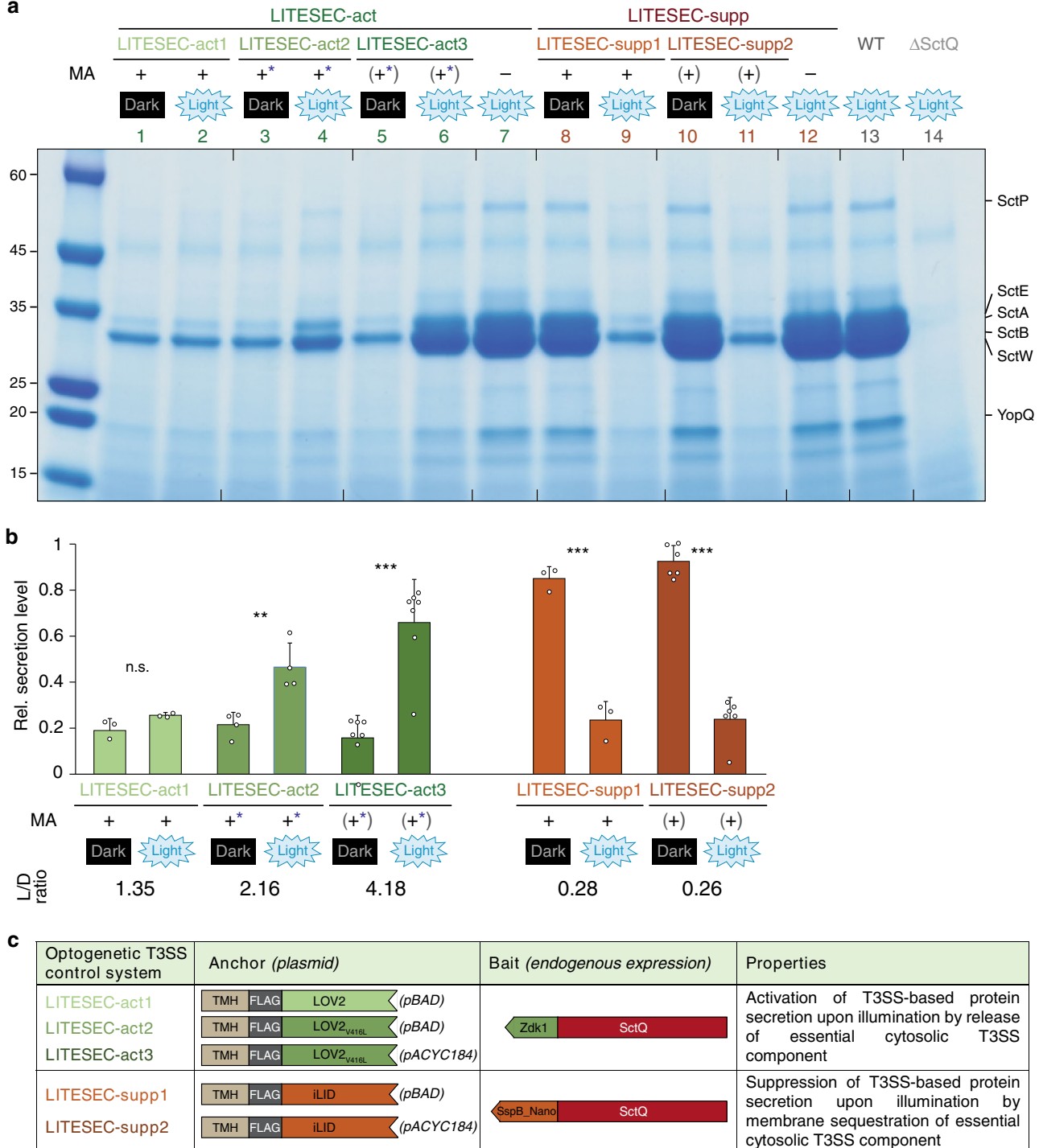

**Fig. 4 Secretion efficiency and light responsiveness in different versions of the LITESEC strains. a** In vitro secretion assay showing light-dependent export of native T3SS substrates (indicated on the right) in various variants of the LITESEC-act strains (lanes 1−7) and LITESEC-supp strains (lanes 8−12), as indicated below. Proteins secreted by $3 \times 10^8$ bacteria during a 180-min incubation period were precipitated and analyzed by SDS-PAGE. Left side, molecular weight in kDa. MA expression level of membrane anchor, + high expression level, (+) low-expression level; − no expression. *, V416L anchor mutant. **b** Quantification of secretion efficiency and light/dark secretion ratio (L/D ratio) for the different LITESEC strains and illuminations indicated above (as in (**a**)). Secretion efficiency was determined by gel densitometry for the SctE/SctA/SctB/SctW bands and normalized for the secretion efficiency in wild-type strains (lane 13 in (**a**)). $N = 3/6/3/4/7$ gel densitometry values from independent experiments for both conditions for LITESEC-act1/2/3/-supp1/2, respectively. Bars show mean values; error bars display standard deviation. Single data points indicated by circles. **$p < 0.01$; ***$p < 0.001$ in a two-tailed homoscedastic $t$ test; n.s. difference not statistically significant (exact values from left to right, $0.08/0.005/4 \times 10^{-5}/4 \times 10^{-4}/5 \times 10^{-8}$). Source data are provided as a Source Data File. **c** Schematic overview of the LITESEC systems and their optogenetic components. All bait proteins are expressed from their native genetic locus. TMH, extended transmembrane helix (see "Methods" for details).

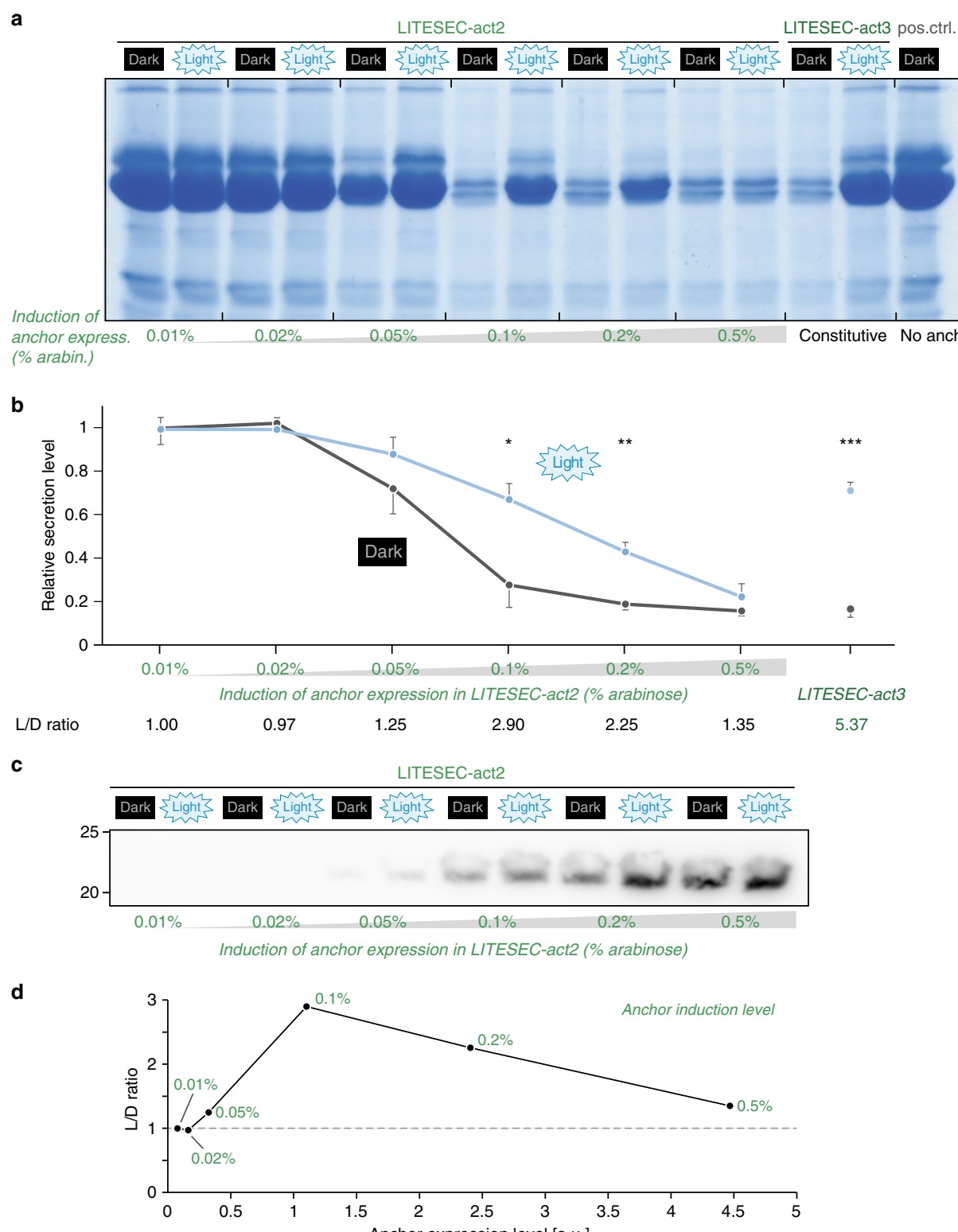

**Fig. 5 The expression ratio of anchor and bait protein dictates the function and light responsiveness of protein secretion in LITESEC-act2. a** In vitro secretion assay showing light-dependent export of native T3SS substrates in the LITESEC-act2 strain at different induction levels of anchor expression. For protein assignment and molecular weight, see Fig. 4a. **b** Quantification of secretion efficiency and light/dark secretion ratio (L/D ratio) for the different expression levels indicated above (as in (**a**)). $N = 3$ gel densitometry values from independent experiments for LITESEC-act2 induced with 0.02 or 0.2% arabinose and LITESEC-act3; $N = 4$ for all other conditions; error bars display standard error of the mean. */**/***$p < 0.05/0.01/0.001$ in a two-tailed homoscedastic $t$ test (exact values from left to right, 0.95/0.67/0.24/0.012/0.009/0.29/3 × $10^{-5}$). **c** Western blot anti-FLAG of total cellular protein of 2 × $10^8$ bacteria in the LITESEC-act2 strain at the indicated induction levels and conditions. Left side, molecular weight in kDa. See Supplementary Fig. 6 for corresponding Ponceau staining as a loading control. **d** Correlation between light/dark secretion ratio (L/D ratio) as determined in (**b**) and anchor expression level. Labels indicate anchor induction levels (arabinose concentrations for LITESEC-act2); the gray dashed line denotes an L/D ratio of 1, indicating light-independent secretion. Source data for panels (**b**–**d**) are provided as a Source Data File.

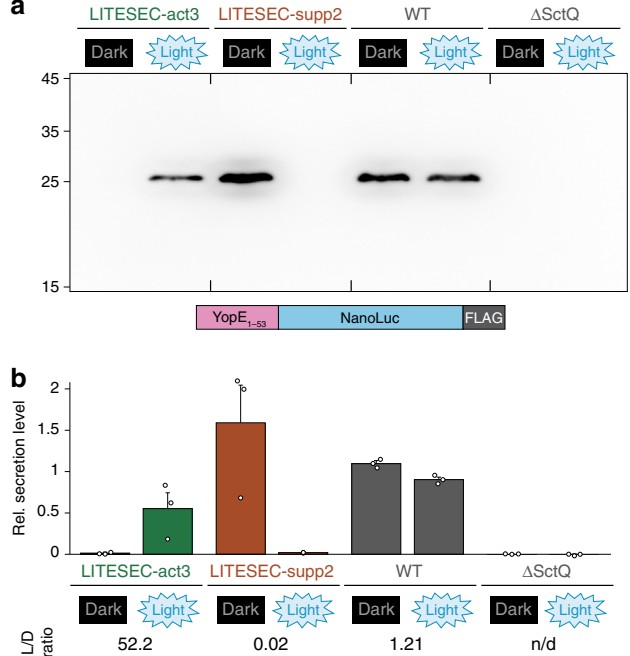

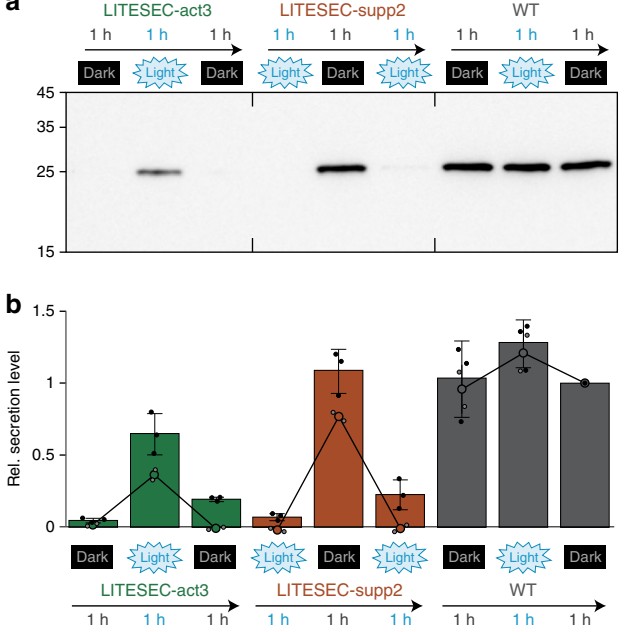

**Fig. 6 Heterologous cargo can be exported in a light-dependent manner.** **a** In vitro secretion assay showing light-dependent export of YopE$_{1-53}$-NanoLuc-FLAG (see scheme below; exp. size, 28.7 kDa), in the indicated strains. Western blot using anti-FLAG antibodies for the culture supernatant of $3 \times 10^8$ bacteria per strain. Left side, molecular weight in kDa. **b** Quantification of light-dependent YopE$_{1-53}$-NanoLuc-FLAG export by densitometric analysis of Western blots, normalized by average secretion of the wild-type control (WT), $N = 3$ densitometry values from independent experiments. Single data points indicated by circles, error bars display the standard error of the mean. L/D ratio, ratio of secretion under light and dark conditions. Source data are provided as a Source Data File.

LITESEC-act3 strain showed an increase of secretion activity over 20 min in light conditions, and required 12−16 min to shut down secretion in the dark (Supplementary Fig. 8).

**Light-induced protein translocation into eukaryotic host cells.** Having found that secretion of heterologous T3SS substrates can be tightly controlled by the LITESEC system, we wanted to employ the LITESEC-act system to control the injection of a cargo protein, YopE$_{1-53}$-β-lactamase, into eukaryotic host cells upon illumination. Translocation of β-lactamase can be visualized by the cleavage of a Förster resonance energy transfer (FRET) reporter substrate, CCF2, within host cells[43,44], which results in a green to blue shift in the emission wavelength. To quantify the light-dependent translocation of the T3SS substrate, we analyzed 671−2694 host cells per bacterial strain and condition. As expected, a wild-type strain translocated the YopE$_{1-53}$-β-lactamase reporter substrate into a high fraction of host cells irrespective of the illumination. The negative control, the same strain expressing the β-lactamase reporter without a secretion signal, displayed a significantly lower rate of blue fluorescence (Fig. 8a), showing that translocation was T3SS-dependent. The LITESEC-act3 strain translocated the transporter in a light-dependent manner, leading to a significantly higher fraction of translocation-positive host cells in light than in dark conditions (close to the positive and negative controls, respectively; Fig. 8b; Supplementary Fig. 9). In contrast, the LITESEC-supp2 strain showed the opposite behavior (Fig. 8b; Supplementary Fig. 9). There was no

**Fig. 7 Secretion of effector proteins can be controlled by light over time.** **a** Export of the heterologous substrate YopE$_{1-53}$-NanoLuc-FLAG in the indicated strains. Secretion-competent bacteria were subsequently incubated under inactivating, activating and inactivating light conditions for 60 min each, as indicated, and the supernatant of $3 \times 10^8$ bacteria was analyzed. Left side, molecular weight in kDa. **b** Quantification of the relative export efficiency (normalized to the wild-type level in the third incubation period) of the strains and conditions shown (**a**) for export of YopE$_{1-53}$-NanoLuc-FLAG as determined by chemiluminescence (bars; $N = 3$ densitometry values from independent experiments, single data points shown as black dots) and immunoblot (circles, continuous line; $N = 2$, single data points shown as gray points); bars show mean values; error bars denote standard deviation. Values for a negative control (ΔSctQ expressing YopE$_{1-53}$-NanoLuc-FLAG) were too small to display in the NanoLuc chemiluminescence assay (<0.001 for all time points). Source data are provided as a Source Data File.

visible reaction of host cells to incubation with T3SS-inactive bacteria, even after extended incubation times (Supplementary Fig. 10), indicating little T3SS-independent effects of bacteria on the used eukaryotic cells. Taken together, these results confirm that translocation of heterologous proteins into eukaryotic host cells by the T3SS can be controlled by external light.

**Light-induced induction of apoptosis in eukaryotic cells.** To directly apply these findings, we established a protocol for the light-controlled induction of apoptosis in host cells, using the proapoptotic protein truncated human BH3 interacting-domain death agonist (tBID) as a T3SS substrate (YopE$_{1-138}$-tBID)[15] in the LITESEC strain background. As controls, we used wild-type bacteria and the T3SS-deficient ΔSctQ strain expressing the same plasmid. Strikingly, strong apoptosis was induced within 1 h after infection, specifically in the HEp-2 cells incubated with bacteria of the LITESEC-act3 strain under light conditions, the LITESEC-supp2 strain under dark conditions, and the positive control (irrespective of the illumination) (Fig. 8c, d). To specifically test for the induction of apoptosis, we detected the apoptosis marker Poly (ADP-ribose) polymerase (PARP) by Western blot in the host cells (Fig. 8e). In addition, we tested the spatial resolution of the activation of the LITESEC-act3 by partially blocking light

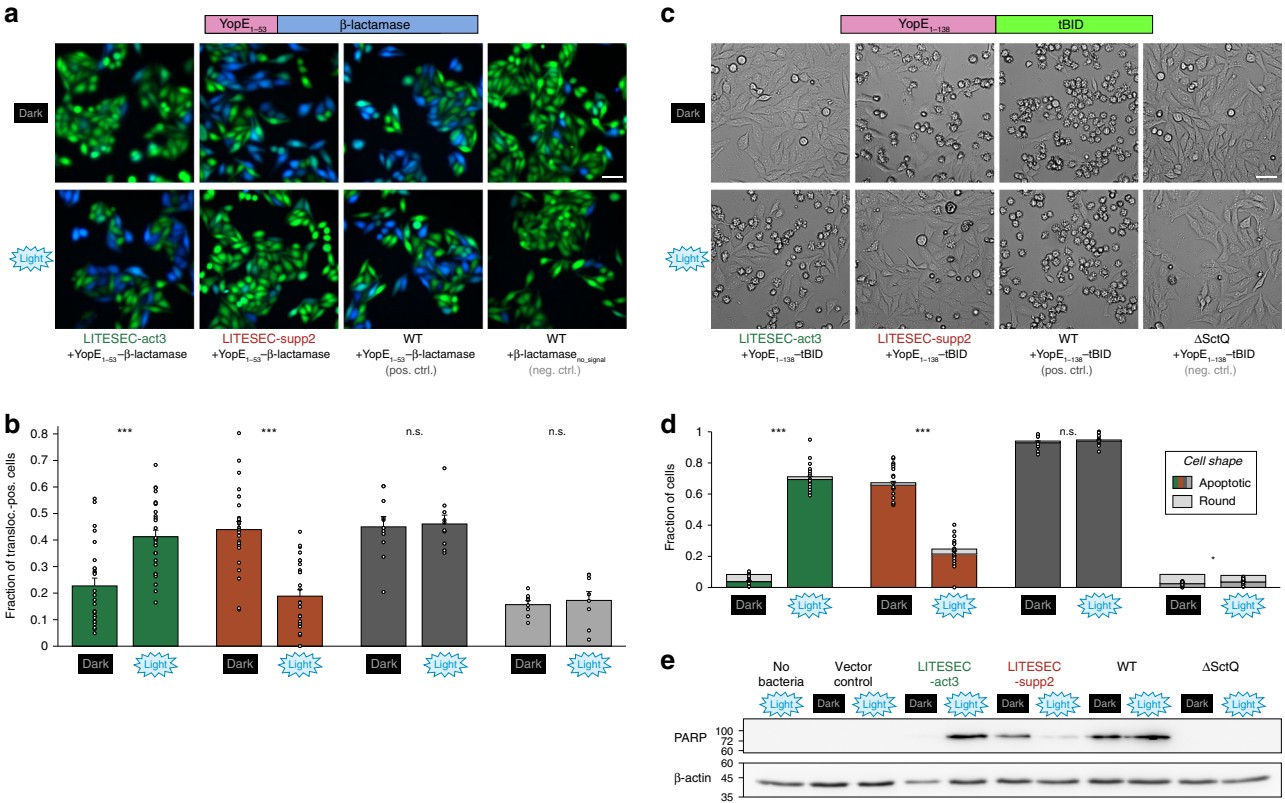

**Fig. 8 Light-dependent translocation of heterologous cargo into eukaryotic host cells. a** Fluorescence micrographs depicting cultured HEp-2 cells that were incubated with the indicated strains expressing either a heterologous T3SS substrate, YopE$_{1-53}$-β-lactamase, or β-lactamase without a secretion signal as a negative control, for 60 min. Translocation of β-lactamase is detected by cleavage of the intracellular β-lactamase substrate CCF2 (leading to loss of FRET, and a transition from green to blue fluorescence emission). Scale bar, 50 μm. **b** Fraction of β-lactamase-positive HEp-2 cells in (**a**) (blue fluorescence). 2343/2423/2226/2694 cells from 26/28/25/27 fields of view from three independent experiments were analyzed for the LITESEC strains under the given conditions from left to right (809/671/995/823 cells from 8/8/10/9 fields of view from three independent experiments for the controls). Single data points (percentage of positive cells per field of view) indicated by circles; error bars display the standard error of the mean. ***$p < 0.001$ in a two-tailed homoscedastic $t$ test; n.s., difference not statistically significant (exact values from left to right, $6 \times 10^{-6}/2 \times 10^{-8}/0.80/0.65$). **c** Micrographs depicting cultured HEp-2 cells that have been incubated with the indicated strains expressing a heterologous T3SS substrate, YopE$_{1-138}$-tBID[15] for 60 min. Translocation of tBID induces apoptosis, which leads to a condensed star-shaped host cell morphology. Scale bar, 50 μm. **d** Visual classification of HEp-2 cells used in (**c**) after infection. 1522/1914/1510/1600/2299/1218/1468/1194 cells from 17/18/17/19/14/13/14/12 fields of view from five independent experiments were analyzed per strain and condition (from left to right). Single data points (percentage of apoptotic cells per field of view) indicated by circles; error bars display the standard error of the mean among fields of view. */***$p < 0.05/0.001$ in a two-tailed homoscedastic $t$ test; n.s., difference not statistically significant (exact values from left to right, $2 \times 10^{-25}/1 \times 10^{-15}/0.40/0.038$). **e** Translocation of tBID induces cleavage of poly (ADP-ribose) polymerase (PARP), which was monitored by Western blot analysis of HEp-2 cells used in (**c**). β-actin was used as a loading control. Left, molecular weigth in kDa. Source data for panels (**b**, **d**, **e**) are provided as a Source Data File.

access to an infection plate, and imaging the host cells afterwards. In line with previous results (Fig. 8c, d, Supplementary Fig. 5), apoptosis was strongly induced in infected cells in the illuminated area and a small (0.25 mm) intermediary region likely to have received some diffracted light (Supplementary Fig. 11). In summary, these results provide a clear example for the application potential of the LITESEC system in cell biology and biotechnology.

## Discussion

To overcome the lack of specificity of T3SS-dependent protein secretion and translocation into eukaryotic cells, we aimed to control T3SS activity by external light. Our solution exploits the recently uncovered dynamic exchange of various essential T3SS components between an injectisome-bound state and a freely diffusing cytosolic state[26,32] to control T3SS-dependent protein secretion by protein sequestration. SctQ, an essential and

dynamic cytosolic component of the T3SS[32], was genetically fused to one interaction domain of two optogenetic sequestration systems, the iLID and LOVTRAP systems[36,37,45], while the membrane-bound interaction domain was coexpressed *in trans*. The two versions of the resulting LITESEC-T3SS system (Light-induced translocation of effectors through sequestration of endogenous components of the T3SS) can be applied in opposite directions: in the LITESEC-supp system, protein export is suppressed by blue light illumination; the LITESEC-act system allows to activate secretion by blue light.

The LITESEC-supp1 system, which is based on the iLID optogenetic interaction switch[35] (Fig. 4c), showed a significant reaction to light (light/dark secretion ratio of 0.28; 24% vs. 85% of wild-type secretion under light and dark conditions, respectively; Fig. 3). Expression of the membrane anchor from a constitutively active promoter on a low-expression plasmid, pACYC184 (LITESEC-supp2) retained the light/dark secretion ratio (L/D ratio of 0.26; 24% vs. 93% WT secretion; Fig. 4), with the

additional advantage that expression of the membrane anchor is constitutive.

For many applications, activation of T3SS protein export upon illumination is preferable. The optogenetic interaction switch employed for this purpose must (i) dissociate upon illumination (allowing activation of the T3SS); (ii) be independent of cofactors that are not present in wild-type bacteria; (iii) be activated by visible or infrared light (compatibility with eukaryotic host cells); (iv) consist of a heterodimer (otherwise, the interaction between individual SctQ molecules, which oligomerize in vivo, would be influenced, which would impact protein secretion[32]). Of the available protein optogenetic interaction switches[38,46,47], the LOVTRAP system[36] fulfills all requirements. However, the first version of the resulting T3SS (LITESEC-act1) only achieved weak activation of T3SS secretion upon illumination (Fig. 4). LITESEC-act2, which uses the V416L mutation in the anchor protein[48] to decrease the affinity between anchor and bait, could be activated by light more efficiently. Even more strikingly, LITESEC-act3, featuring a reduced expression level of the V416L variant of the membrane anchor, led to a strong activation of T3SS protein secretion upon illumination, while retaining the tight suppression of secretion in the dark (L/D ratio of 4.2; 66% vs. 16%; Fig. 4). In the LITESEC-supp system, the decreased anchor levels in the -supp2 variant did not lead to significant changes in the activation of secretion, suggesting that this system is less sensitive to changes in the anchor/bait ratio, and indicating that the achieved L/D ratio is the optimum for native secretion substrates. For a more detailed discussion of activation dynamics and applications of the LITESEC system, see Supplementary Discussion.

Notably, the export of heterologous cargo was entirely light-dependent (no visible export under inactive conditions; Fig. 6) in both LITESEC variants. The LITESEC-act3 system was less efficient for heterologous cargo expressed from plasmid (around 50% secretion efficiency) and activated more slowly than LITESEC-supp2 (Figs. 6, 7, Supplementary Fig. 8). This indicates that despite the V416L point mutation (LITESEC-act2) and the changed anchor/bait expression ratio, parts of SctQ remain tethered to the membrane after illumination. For the majority of applications, the tight regulation of secretion in the off state shared by both LITESEC systems will be more important than the maximal translocation rate. Interestingly, endogenous T3SS translocator proteins were still secreted to a basal level under inactivating light conditions, even in the most tightly controlled strains (LITESEC-act3/-supp2; Fig. 4). This might indicate that the export of heterologous cargo is regulated differently from the export of the endogenous translocators, which for example also involves protein-specific chaperones. While this hypothesis remains to be rigorously tested, it highlights that beyond their application, LITESEC and similar optogenetic approaches can help to better understand the underlying biological systems.

To explore the influence of the anchor/bait expression ratio on light control of the T3SS in more detail, we measured the light-dependent activation of the LITESEC-act2 system at different expression levels of the anchor protein. The results indicate that intermediate anchor/bait ratios allowed an optimal response to blue light for the LITESEC-act system. Higher ratios retain partial membrane sequestration under light conditions and subsequently impair T3SS activity in the activated stage; conversely, low ratios lead to incomplete sequestration and measurable T3SS activity under nonactivating conditions (Fig. 5). Taken together, our data strongly suggest a relatively tight sweet spot in the expression ratio of the two interacting proteins, which may be key for the successful optogenetic control of bacterial processes. This is in contrast to the eukaryotic application of the LOVTRAP interaction switches where high anchor/bait concentrations were shown to be optimal[36]. We therefore propose that optimization of the

anchor/bait expression ratio represents an important step in the design of optogenetically controlled processes in prokaryotes.

The successful development and application of the LITESEC system highlights some key features for the control of intracellular processes in prokaryotes by optogenetic interaction switches. The target protein (in our case the essential T3SS component SctQ) (i) has to be functional as a fusion protein to an optogenetic interaction domain, (ii) must be present in the cytosol at least temporarily to allow sequestration to occur, and (iii) must not be functional when tethered to the membrane anchor protein. To fulfill the last criterion, the target protein may feature (a) a specific place of action (such as the injectisome for SctQ in the present case), or (b) a specific interaction interface that is rendered inaccessible by the interaction with the anchor. In eukaryotic systems, proteins have been sequestered to various structures including the plasma membrane or mitochondria. The simpler cellular organization of bacteria makes the inner membrane an obvious target for protein sequestration, to which interaction domains can be easily targeted to by the addition of N-terminal TMHs. While the nature of the TMH is likely to be secondary for the success of the application, the extended TatA TMH and a short glycine-rich linker worked well for our approach. Crucially, we found that the expression ratio between anchor and bait proteins is a key determinant for the success of LITESEC and, most, likely, similar approaches to control bacterial processes by light.

The T3SS is a very promising tool for protein delivery into eukaryotic cells, both in cell culture and in healthcare[11,15,21]. However, the T3SS indiscriminately injects cargo proteins into contacting host cells. Lack of targetability is therefore a main obstacle in the further development and application of this method[21,22]. Previous methods to control the activity of the T3SS relied on controlled expression of one or all components of the injectisome. For example, Song et al.[49] expressed all components of the *Salmonella* SPI-1 T3SS from two inducible promoters in a clean expression system, and Schulte et al.[50] expressed the T3SS genes from a TetA promoter, which additionally allows the intracellular induction of the T3SS. Besides the difficulty to specifically induce secretion in defined places in situ, the main drawback of these methods is the slow response (induction of expression and assembly of the T3SS take more than 60 min[29,49,50]). In addition, in these systems, the T3SS remains active as long as the induced protein(s) are still present, which leads to a higher risk of translocation into nontarget cells.

By using light to specifically activate the modified T3SS in bacteria, we have addressed this issue. The LITESEC system allows to deliver proteins into host cells at a specific time and place. The system gives complete control over the secretion of heterologous T3SS cargo into the supernatant, either by providing illumination (LITESEC-act), or stopping the light exposure (LITESEC-supp). Importantly, secretion by the LITESEC-act system is temporary, and stopped within minutes after the end of illumination with blue light, thereby further reducing unspecific activation. An important consideration in the application of bacteria for protein translocation into eukaryotic cells is a possible toxic or immunogenic effect of the bacterial vector[21,22]. The *Y. enterocolitica* strain used in this study has been depleted of the main virulence effectors YopH,O,P,E,M,T and is a cell wall synthesis auxotroph, which cannot replicate in the absence of externally added diaminopimelic acid. Accordingly, LITESEC bacteria in which secretion is not activated have little or no visual effect on the host cells, even after long incubation of the cells after infection (Supplementary Fig. 10). However, the ongoing development of less immunogenic bacterial vectors through modification or deletion of pathogen-associated molecular patterns and other means[11,21,51,52] remains an essential factor for the

successful application of this technique in medicine and biotechnology.

The LITESEC system presented in this work uses light-controlled sequestration of an essential dynamic T3SS component to precisely regulate the activity of the T3SS. This approach provides a new method for highly time- and space-resolved protein secretion and delivery into eukaryotic cells.

## Methods

**Plasmids.** Plasmids and strains used in this study are listed in Supplementary Tables 1 and 2, respectively.

Expression plasmids were created using Phusion polymerase (New England Biolabs), delivered into *E. coli Top10* by electrotransformation, and confirmed by sequencing (Eurofins Genomics). Mutator plasmids were created by flanking the gene of interest with wt gene regions to allow an allelic exchange, using the suicide plasmid pKNG101[39]. Mutator plasmids were propagated in *E. coli Sm10 λpir+* and allelic exchange was performed by two-step homologous recombination[39].

**Cultivation of bacteria.** *Y. enterocolitica* strains were cultivated in rich BHI (Brain Heart Infusion Broth) medium (3.7% w/v), complemented with nalidixic acid (35 μg ml$^{-1}$) and 2,6-diaminopimelic acid (DAP, 60 μg ml$^{-1}$) (cultivation medium). *E. coli* strains were cultivated in LB (Lurea Broth) medium (tryptone (10% w/v), yeast extract (5% w/v), NaCl (10% w/v)). Where required, the medium was supplemented with ampicillin or carbenicillin (200 μg ml$^{-1}$), chloramphenicol (25 μg ml$^{-1}$), or streptomycin (50 μg ml$^{-1}$). For overnight cultures, 2−5 ml of cultivation media with corresponding antibiotics were inoculated and cultivated overnight at 28 °C (*Y. enterocolitica*) or 37 °C (*E. coli*) in a shaking incubator.

**T3SS in vitro secretion assay.** Day cultures were inoculated from stationary overnight cultures (1:50 dilution for nonsecreting conditions, 1:41.67 for secreting conditions) in cultivation medium additionally complemented with MgCl$_2$ (20 mM), glycerol (0.4% w/v), and corresponding antibiotics. CaCl$_2$ (5 mM) or EGTA (5 mM) were added for nonsecreting and secreting conditions, respectively. The cultures were cultivated for 90 min at 28 °C and then shifted to a 37 °C water bath and incubated for 2−3 h (unless stated otherwise, plasmids were induced with 0.2% w/v L-arabinose before shifting to 37 °C). The optical density of cultures at 600 nm was determined. Two milliliters of bacteria were collected by centrifugation, and proteins from 1800 μl supernatant were precipitated with 200 μl trichloroacetic acid for at least 1 h at 4 °C. The precipitated proteins were collected by centrifugation, washed with ice-cold acetone, and resuspended in SDS-PAGE loading buffer (SDS (2% w/v), Tris (0.1 M), glycerol (10% w/v), dithiothreitol (0.05 M), pH = 6.8) to normalize (15 μl/3 × 10$^8$ bacteria). Unless stated differently, proteins secreted by an equivalent of 3 × 10$^8$ bacteria were loaded onto SDS-PAGE gels.

**Immunoblotting and immunodetection.** Two milliliters of bacterial culture was collected and the total cell pellet was resuspended with an equivalent amount of SDS-PAGE loading buffer to normalize (15 μl/1.5 × 10$^8$ or 2 × 10$^8$ bacteria). Total cellular protein (TC) or supernatant (SN) samples were heated at 99 °C for 5 min before loading on an SDS-PAGE gel.

SDS-PAGE gels were run for 1−1.5 h (135 V, 400 mA), using BlueClassic Prestained Marker (Jens Biosciences) or Precision Plus Protein Standard #1610373 (Bio-Rad) as size standards, and blotted on an Amersham nitrocellulose membrane (0.2 μm) using a Turbo-Blot Transfer-system with the settings: 1.3 A, 25 V, 7 min. Immunoblots were carried out using primary rabbit antibodies against *Y. enterocolitica* SctQ (MIPA235, 1:1000), mCherry (Biovision 5993 BV-F7G, 1:2000), the Flag peptide (Rockland 600-401-383, 1:2000), cleaved PARP (Asp 214) (Cell Signaling Technology 5625, 1:500), or mouse antibodies against β-actin (Santa Cruz sc47778, 1:200) in combination with secondary antibodies goat anti-rabbit or sheep anti-mouse, both conjugated to horseradish peroxidase (Sigma A8275 and Amersham NXA931, 1:5000), and visualized using ECL chemiluminescent substrate (Pierce) on a LAS-4000 Luminescence Image Analyzer.

**Fluorescence microscopy.** For fluorescence microscopy, strains were cultivated as described above under nonsecreting conditions. Two milliliters of culture then was harvested for 4 min at 2400 × *g* and the cell pellet was resuspended in 400 μl of minimal medium (4-(2-hydroxyethyl)-1-piperazineethanesulfonic acid (HEPES) (100 mM), (NH$_4$)$_2$SO$_4$ (5 mM), NaCl (100 mM), sodium glutamate (20 mM), MgCl$_2$ (10 mM), K$_2$SO$_4$ (5 mM), casamino acids (0.5% w/v)) including DAP (60 μg ml$^{-1}$). Two microliters of the resuspension was spotted onto agar slides (1.5% w/v agarose in minimal medium) and topped with a circular cover slip (25 mm diameter). Samples were analyzed on an inverse fluorescence microscope (Deltavision Elite). For pulse activation of the optogenetic interaction switches, 0.1 s of GFP excitation light (~480 nm, light intensity ~ 2.5 mW cm$^{-2}$) was applied. Unless stated differently, exposure times were 500 ms for mCherry fluorescence, using a mCherry filter set (575/25 excitation and 625/45 nm emission filter sets), and 200 ms for GFP fluorescence, using a GFP filter set (475/28 and 525/48 nm, respectively). Per image, a *z* stack containing 7−15 frames per wavelength with a

spacing of 150 nm was acquired. Micrographs were processed using softWoRx 7.0.0, and the ImageJ 1.52i-based Fiji software package[53,54] was used for image analysis and display.

**Optogenetic cell cultivation.** For optogenetic experiments, the strains for secretion assays or Western blots (to determine the amount of secreted proteins) were cultivated under secreting conditions. At the indicated time points after induction of the system by a temperature shift to 37 °C, the cultures were cultivated at 37 °C for 1−3 h in an optogenetic experimental setup in a shaking incubator, consisting of two blue light sources that were placed around the bacterial cultures (light source 1, "Globo lighting 10 W LED 9 V 34118 S", Globo Lighting GmbH; light source 2, "Rolux LED-Leiste DF-7024-12 V 1.5 W", Rolux Leuchten GmbH). Bacteria were cultivated at 37 °C under blue light or dark conditions (light intensity at culture location at a wavelength of 488 nm was ~1 mW cm$^{-2}$), and further processed as described.

**Infection assays.** HEp-2 cells were maintained in Roswell Park Memorial Institute (RPMI) 1640 medium (Gibco) supplemented with 7.5% newborn calf serum (Sigma-Aldrich) in 5% CO$_2$ at 37 °C. For the β-lactamase translocation assay, the infection assay was adapted from ref. [55]. HEp-2 cells were seeded into Nuncion Delta Surface 96-well flat plates (Thermo Scientific) at a cell density of 2.0 × 10$^4$ cells per well. Prior to infection, 5 mM DAP was added to the medium of the seeded HEp-2 cells. Day cultures were inoculated from stationary overnight cultures (1:25 dilution) in BHI supplemented with DAP (60 μg ml$^{-1}$), MgCl$_2$ (20 mM), and glycerol (0.4% w/v). Expression of the cargo protein from the pBAD plasmid was induced with 0.2% arabinose (w/v), unless stated differently. The cultures were incubated for 90 min at 37 °C under activating conditions (dark for LITESEC-supp/light for LITESEC-act) to induce T3SS formation. After incubation, the bacteria were centrifuged for 4 min at 4500 × *g* and 4 °C. The bacteria were resuspended in ice-cold phosphate buffered saline (PBS) containing DAP (60 μg ml$^{-1}$) to a density of approximately 2.5 × 10$^8$ cfu ml$^{-1}$, incubated on ice in "off" conditions (light for LITESEC-supp/dark for LITESEC-act) for 15 min, then added to a semiconfluent layer of HEp2-cells at a multiplicity of infection (MOI) of approximately 100, and incubated under blue light or dark conditions for 60 min at 37 °C in 5% CO$_2$. Following incubation, the cell culture medium was removed and 100 μl of working solution were added (1:3 dilution RPMI 1640 medium without phenol red (Gibco) in PBS (Gibco) with 25 mM probenecid acid (Alfa Aesar) dissolved in cell culture grade dimethyl sulfoxide (DMSO) (Santa Cruz)). For β-lactamase translocation, 20 μl of CCF2-AM were added (0.12 μl solution A, 1.2 μl solution B and 18.68 μl solution C (solutions A, B and C provided from Invitrogen CCF2-AM loading kit)). After 5 min of incubation, the working solution and CCF2-AM were removed and 100 μl of fresh working solution was added. Plates were then incubated at 37 °C in 5% CO$_2$ for 10 min. Next, cells were fixed by addition of 100 μl of ice-cold 1% paraformaldehyde (PFA) (w/v) and incubation on ice for 10 min. As a last step, the PFA solution was replaced by PBS. Fields of view were chosen in the differential interference contrast (DIC) channel, preventing any bias, and all fields of view were analyzed. Translocation of YopE$_{1−53}$-β-lactamase was detected by comparing the fluorescence emission at 525/48 nm (FRET-based emission of uncleaved CCF2) vs. 435/48 nm (emission of cleaved CCF2, equivalent to substrate translocation), both at an excitation at 390/18 nm. Both channels were background-corrected. For the apoptosis assay, the protocol outlined in ref. [15] was adapted as follows. HEp-2 cells were seeded to a density of 1.18 × 10$^5$ cells per well into Nuncion Delta Surface 24-well plate (Thermo Scientific). Prior to infection, DAP (60 μg ml$^{-1}$) and 0.2% arabinose (w/v) were added to each well of HEp-2 cells. Bacteria were first grown at 28 °C for 90 min and then shifted to 37 °C for 120 min, collected by centrifugation, and resuspended in PBS (prewarmed to 37 °C) containing 5 mM DAP at a density of approximately 2.5 × 10$^8$ cfu ml$^{-1}$. The bacteria were added to a semiconfluent layer of HEp2-cells at an MOI of approximately 140, and incubated under blue light or dark conditions for 60 min at 37 °C in 5% CO$_2$. Following incubation, the cell culture medium was removed and 300 μl of RPMI medium (GIBCO 1640) containing gentamycin (100 μg ml$^{-1}$) was added. Cells were incubated for further 60 min at 37 °C in 5% CO$_2$ under the specified light conditions and were then imaged with a binocular microscope (×5 objective) or on an inverse fluorescence microscope (Deltavision Elite) (×20 objective). Fluorescence and cell shape of HEp-2 cells were manually classified by blinded observers.

**Luciferase assay.** Day cultures (20 ml) were inoculated from stationary overnight cultures (1:50 dilution) in nonsecreting cultivation medium, as described above, and incubated for 90 min at 28 °C. Subsequently, expression of the YopE$_{1−53}$-NanoLuc-Flag cargo protein from the pBAD plasmid was induced with 0.2% arabinose (w/v). The cultures were incubated for 120 min at 37 °C under activating conditions (dark for LITESEC-supp/light for LITESEC-act) to induce T3SS formation. After that, strains were incubated for 10 min under dark conditions, and 5 mM EGTA was added to start secretion. Bacteria were then incubated for 60 min each (20 min each for the determination of switching kinetics) at the indicated conditions. Samples were taken immediately before EGTA addition and at the indicated times afterwards. In the samples, 10 mM CaCl$_2$ was added to stop secretion. Bacteria were harvested and the supernatant was used for the enzymatic assay. The enzymatic Nanoluc detection assay was performed according to the

manufacturer instructions, similar to ref. [42]. Five microliters of supernatant was mixed with 25 μl H$_2$O and 30 μl of NanoLuc detection reagent (Nano-Glo Luciferase Assay Substrate in Luciferase Assay Buffer, PROMEGA Corporation, Madison). Bioluminescence was detected in an Elisa Plate Reader Infinite M20 Pro (BioTek Instruments, Vermont), smallest field of view, large binning and an aquisition time of 1000 ms.

**Statistics and reproducibility**. For the experiments that were quantified, the number of replicates is indicated in the respective figure legends. For the experiments where representative images are shown, the experiments were repeated independently with similar results for the following number of times: Fig. 3, 3 times (>10 times for the strains with membrane anchors and the controls); Fig. 5c, 3 times; Fig. 8e, 4 times; Supplementary Fig. 1C, 3 times; Supplementary Fig. 2A, 3 times; Supplementary Fig. 2B, one Ponceau stained gel, normalization of the amount loaded on the gel in every case (N = 3); Supplementary Fig. 3A, >10 times; Supplementary Fig. 5, once (the influence of ambient light was observed with similar outcome in other experimental settings >10 times); Supplementary Fig. 6, one Ponceau stained gel, normalization of the amount loaded on the gel in every case (N = 3); Supplementary Fig. 10, >10 fields of view in two independent experiments with consistent results across different time points after infection; Supplementary Fig. 11A, B, 5 wells on two plates from two independent experiments with different light-intransparent shapes.

**Reporting summary**. Further information on research design is available in the Nature Research Reporting Summary linked to this article.

## Data availability
All data generated or analyzed during this study are included in this published article and its supplementary information files. Source data for Figs. 2c, d, 4b, 5b–d, 6b, 7b, 8b, d, e and Supplementary Figs. 2, 3, 7, 8, 9, 11 are provided as a Source Data File.

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

## Acknowledgements

The authors thank Prof. Andreas Gahlmann, University of Virginia, for important contributions and discussions during the inception of this study; Prof. Klaus Hahn, University of North Carolina, for the LOVTRAP system templates (Addgene 81010, 81036, 81041); Prof. Brian Kuhlman, University of North Carolina, for the iLID system templates (Addgene 60408, 60409); Dr. Felicity Alcock, University of Oxford, for suggestions for the design of the transmembrane anchor; Dr. Seraphine Wegner, Max Planck Institute for Polymer Research, Mainz, for helpful discussions about optogenetic interaction switches; Prof. Petra Dersch, University of Münster, for the gift of HEp-2 cells; Dr. Simon Ittig, T3 Pharmaceuticals AG, Basel, for the kind provision of the plasmid encoding YopE$_{1-138}$-tBID; Prof. Lars-Oliver Essen, University of Marburg, for input and support on the characterization of the optogenetic setup; Horst Henseling, Max Planck Institute for Terrestrial Microbiology, Marburg, for the design and construction of microscopy chambers; Dr. Cornelia Kilchert, University of Gießen, for the provision of the beta-actin antibody; Dr. Oleg Timofeev, University of Marburg, for help with the apoptosis assays; and Dr. Timo Glatter, Max Planck Institute for Terrestrial Microbiology, Marburg, for useful discussions.

## Author contributions

F.L. created and characterized the LITESEC strains and performed the majority of the experiments and data analysis. B.M.-D. established the infections assays and participated in experiments and data analysis. K.L. provided the eukaryotic cell culture and participated in infection assays. T.S. provided methodological input and reagents for analysis of apoptosis. A.D. conceived the study and experimental setup, participated in data analysis and wrote the manuscript.

## Competing interests

F.L. and A.D. declare the following competing interests: A European patent application and PCT registration on the presented method was filed by the Max-Planck-Gesellschaft zur Förderung der Wissenschaften e.V.: Lindner, F., Gahlmann, A., Diepold, A. "Optogenetic control of protein translocation—Protein secretion and translocation into eukaryotic cells with high spatial and temporal resolution by light-controlled activation of the bacterial type III secretion system", European Patent Application 19166308, March 2019 and PCT registration, March 2020. The remaining authors declare no competing interests.
