## [Peer Review File · Nature Communications]

Reviewers' comments:

Reviewer #1 (Remarks to the Author):

In this manuscript, authors created a LITESEC(Light-induced translocation of effectors through sequestration of endogenous components of the T3SS) system using T3SS, a component of bacterium that translocate proteins into eukaryotic host cells. Of the cytosolic components of T3SS, sctQ, which is known to dynamically regulate protein secretion between the cytosol and the injectisome, is combined with LOV or iLID to enable protein secretion and secretion suppression optogenetically. The authors used LOV domain to create LITESEC-act version that secretes effector proteins in the light but not in the dark. LITESEC-supp version used the iLID that secretes in the dark but not in the light. The LITESEC system has the advantage of its reversible characteristics for activation. Normal translocation system is known for having low translocation rates for stable and large proteins, but T3SS allows for the rapid and efficient release of most cargos and can be applied to even eukaryotic cells. This advantage has potential to apply in various ways to further research. However, there are still points to overcome such as its difficulty of target specificity.

In conclusion, their approach in the manuscript is not sufficiently characterized to demonstrate the efficiency of LITESEC system. Further experiments and controls should be done to support the conclusions of this study. As it is a tool that can be controlled by light, additional figures are necessarily shown such as imaging data instead of SDS-FAGE without control. I strongly recommend the authors to improve the following issues regarding the characterization of this tool to strengthen their work. If authors are able to exhibit data described to improve their supports to the study, I believe the author can make it a strong candidate.

Major points:

1. When introducing two optogenetic interaction switches in the introduction part, the iLID system uses the SSPB-nano version. Although there are micro, nano, and milli versions in *sspb*, the authors should compare the *sspb* versions and explain why the nano version was used.
2. In the case of LITESEC-supp, the difference is very clear in the light and dark states, but in the case of LITESEC-act, the expression is not clear, unlike LITESEC-supp. The author made the act using the uncaging properties of the LOV domain by light, but it might be a good idea to try another light uncaging protein to improve activation of LITESEC-supp, for example, Dronpa or something else, and see how the results differ from the LOV.
3. Since spatial controllability is the advantage of the optogenetic tool, it is necessary to perform additional spatial experiments such as lighting only a few cells and comparing with other cells or only lighting a point within the same cell.

Minor points:

1. There is no loading control in the figure showing the western result, and there is no size bar in the figure showing the cell.
2. Sup. table 1 says that the unlabeled anchors and anchors match, but the additional explanation is necessary to support this.
3. In Figure 2AB, the dark/light notation would be better to mark the bar as 2CD or outside of the image. In addition, a prominent iLID and LOVTRAP mark is required.
4. In Supple figure 2, the 40.8kDa of *zdk1-sctQ* / 46.7kDa of SSPB nano-sctQ looks larger than 40.8kDa / 46.7kDa and the WT sctQ is described as 34.4kDa but it is larger than 35kDa. The mismatch between the size label and the actual protein size is obviously wrong. The authors need to explain the mismatch of the size and labeling. Also, it is better to use names instead of numbers to recognize them at a glance.
5. When switching from pBAD to pACYC, the expression of the anchor protein changes to low. In the LITESEC system using pACYC, secretion increases in light in *act3* and suppression in *supp2*. This is opposite to the result with the same low copies vector. Could the author explain what this interpretation is?
6. The pACYC184 plasmid is known to have a high copy number, but in supple figure 4, pACYC184 shows a lower expression than pBAD. However, prior to knowing this result of experiment, it was referred to each of pBAD / pACYC184 as medium-high / low copy in figure 4. It may be confusing to say the results

of the experiment later, so it may be good to remove the mention or change the composition of the figures.

7. It is good to highlight LITESEC-act3 in Figure 5B, but it is not good to indicate the size of the circle from act2 or the position of * / ** / *** differently. Also, the L / D ratio between 5B and 5C is about 5B, which is closer to 5C.

8. In the single image of the figure, there is light/dark on the top (ex. Figure 3) and on the bottom (ex. suppl 4). When the images are unified in one position, it would be easier to understand the figure.

9. In figure 7, both act and supp are shown to be reversible as the authors describe the advantages of the LITESEC system. However, it would be better to add the image using fluorescence.

10. When authors see the same system in Figure 8A and the change in the dark and light states, authors should observe the same cells in the same location. As the author noted that there is a basal effect, it is not clear if the imaging position is different and the change is large enough because of the basal effect.

Reviewer #2 (Remarks to the Author):

Lindner et al describe the development of a beautiful method to induce or repress the secretion activity of bacterial type III secretion systems by optogenetic control. While I admire the idea and its development, I miss the presentation of a true proof of concept, a used case. The authors discuss the potential applications of the system in detail, in particular in the supplement, but do not go the one step further, to actually proof that the system would work in a "real" case. Clearly, optogenetic control has its limitations for use in deeper tissues and so the authors mainly suggest its use for cultured cells. I am afraid, however, that non-bacteriologist researchers and technicians will not enjoy working with alive bacteria in their cell culture systems, not to speak of clinical applications. Importantly, the authors do use a bacterial strain that has a much reduced pathogenicity due to the lack of type III-secreted effectors and is hampered in replication because of an *asd* mutation. But still, these bacteria express plenty of MAMPs that will influence signaling pathways inside the targeted eukaryotic cells.

Besides this general critique, the paper is very well written, in its logical build-up as well as in its technical detail. It features very clear figures and figure legends, sufficiently detailed methods (where provided, see below) as well as reasonable statistics where appropriate.

I also feel that this technique has substantial potential as a tool to investigate type III secretion system function - but the authors do not make use of it in the work described to this point.

Minor points:

Fig. 1 shows a very much outdated cartoon of the T3SS with respect to the architecture of the export apparatus and of the cytoplasmic components. It would be nice if the authors update the cartoon appropriately.

Fig. 5C: The labeling of the Western blot image is not very clear. I assume that the labeling of Fig. 5A also applies to Fig. 5C but a repetition of the labeling or some other structuring element might makes this more clear.

Line 228: "inlight" should be "in light", I assume.

Suppl. Methods: I miss the description of the SDS PAGE and Western blot techniques used. Antibodies and their concentrations are not provided, nor the specific detection and quantification methods used. In case the authors use ECL-based immunodetection, I I question the reliability of the quantification provided in the respective figures.

Reviewers' comments:

Reviewer #1 (Remarks to the Author):

In this manuscript, authors created a LITESEC(Light-induced translocation of effectors through sequestration of endogenous components of the T3SS) system using T3SS, a component of bacterium that translocate proteins into eukaryotic host cells. Of the cytosolic components of T3SS, SctQ, which is known to dynamically regulate protein secretion between the cytosol and the injectisome, is combined with LOV or iLID to enable protein secretion and secretion suppression optogenetically. The authors used LOV domain to create LITESEC-act version that secretes effector proteins in the light but not in the dark. LITESEC-supp version used the iLID that secretes in the dark but not in the light. The LITESEC system has the advantage of its reversible characteristics for activation. Normal translocation system is known for having low translocation rates for stable and large proteins, but T3SS allows for the rapid and efficient release of most cargos and can be applied to even eukaryotic cells.

This advantage has potential to apply in various ways to further research. However, there are still points to overcome such as its difficulty of target specificity.

In conclusion, their approach in the manuscript is not sufficiently characterized to demonstrate the efficiency of LITESEC system. Further experiments and controls should be done to support the conclusions of this study. As it is a tool that can be controlled by light, additional figures are necessarily shown such as imaging data instead of SDS-PAGE without control. I strongly recommend the authors to improve the following issues regarding the characterization of this tool to strengthen their work. If authors are able to exhibit data described to improve their supports to the study, I believe the author can make it a strong candidate.

We thank the reviewer for the positive evaluation and the detailed suggestions! We have addressed the specific points as indicated below. For the evaluation of secretion activity, SDS-PAGE gels are the gold standard in the field. As the reviewer rightly points out, these gels do not include a loading control. The reason for this is that there is no suitable control protein that is exported T3SS-independently under the conditions that induce T3SS activity (in the wild-type strain, more than 90% of the proteins in the supernatant are T3SS-dependently exported proteins). To normalize for differences in culture density, all samples are adjusted for the culture OD, so that in each lane, the culture supernatant of 3×10^9 bacteria is loaded. We now mention this fact specifically in the figure legends and methods part, in addition to the biological controls that are included in each figure (wild-type strains as positive controls and deletions of single T3SS components as negative controls).

Major points:

1. When introducing two optogenetic interaction switches in the introduction part, the iLID system uses the SSPB-nano version. Although there are micro, nano, and milli versions in sspB, the authors should compare the sspB versions and explain why the nano version was used.

As mentioned by the reviewer, different versions of the SspB proteins display a range of affinities to the iLID interaction partner and confer different properties to the iLID system, with dissociation constants that span three orders of magnitude (Zimmerman et al., Biochemistry 2016). We estimate that the cellular concentration of our target protein, SctQ, is around 1-2 μM (~ 1000 molecules per bacterium $\approx (1 \mu\text{m})^3$). Given that our system requires efficient membrane tethering in the dark state, we chose the SspB-nano version, which has a dissociation constant of 132 nM in the light and 4.7 μM in the dark, over SspB_micro and SspB_milli, which have roughly 10 and 1000 times higher dissociation constants. We now specifically mention this reasoning in the introduction (lines 124-128 in the manuscript version with highlighted changes).

2. In the case of LITESEC-supp, the difference is very clear in the light and dark states, but in the case of LITESEC-act, the expression is not clear, unlike LITESEC-supp. The author made the act using the uncaging properties of the LOV domain by light, but it might be a good idea to try another light uncaging protein to improve activation of LITESEC-supp, for example, Dronpa or something else, and see how the results differ from the LOV.

This is a good suggestion. However, the requirements for the optogenetic interaction switch that can be used in the LITESEC-act system are very specific. The employed system must fulfil all following requirements:

- Dissociation upon illumination (allowing activation of the T3SS)
- No requirement of cofactors that are not present in wild-type bacteria
- No UV light requirement (compatibility with eukaryotic host cells)
- Heterodimer (otherwise, the interaction between individual SctQ molecules, which oligomerize *in vivo*, would be influenced, which would almost certainly inhibit protein secretion)

To the best of our knowledge, the LOVTRAP system is the only system that fulfils all of these requirements without the need of a cofactor. We now include this reasoning in the discussion (lines 410-416) and additionally discuss the differences in maximal activation level between the LITESEC-act and -supp variants (lines 428-433).

3. Since spatial controllability is the advantage of the optogenetic tool, it is necessary to perform additional spatial experiments such as lighting only a few cells and comparing with other cells or only lighting a point within the same cell.

This is a very valid point. The application of the LITESEC system to induce apoptosis in eukaryotic cells (new Fig. 8C-E) provided a good test case for this. The result showcasing the spatial resolution is now included as Suppl. Fig. 11.

Minor points:

1. There is no loading control in the figure showing the western result, and there is no size bar in the figure showing the cell.

The Western Blots in Fig. 6A and 7A display the specific detection of heterologous cargo in the supernatant. As discussed earlier, there is no protein that could be used for a loading control in the supernatant. Instead, these samples are normalized by optical density of the corresponding bacterial culture at the time the supernatant samples were taken, which is now specifically mentioned in the figure legend. The Western Blot in Fig. 5C shows the expression of FLAG-tagged anchor proteins. In this case, we have now added the Ponceau staining of the membrane that can serve as a loading control in a new Supplementary Figure (Suppl. Fig. 6). In Fig. 2, we now specifically mention the size of the scale bar and the enlargement factor of the enlarged insets.

2. Sup. table 1 says that the unlabeled anchors and anchors match, but the additional explanation is necessary to support this.

Both Table 1 and Suppl. Table 1 described the constructs used in different parts of the manuscript. We have now added a visual representation of the constructs used in the main parts of the manuscript (previously Table 1, new Fig. 4C), and additionally visualize the construct used in Suppl. Fig. 1 in the figure itself.

3. In Figure 2AB, the dark/light notation would be better to mark the bar as 2CD or outside of the image. In addition, a prominent iLID and LOVTRAP mark is required.

We thank the reviewer for this suggestion; we now have moved the light/dark labels outside the micrograph, and have included prominent iLID / LOVTRAP labels to make the figure more easily understandable.

4. In Supple figure 2, the 40.8kDa of zdk1-sctQ/ 46.7kDa of SSPB nano-sctQ looks larger than 40.8kDa/ 46.7kDa and the WT sctQ is described as 34.4kDa but it is larger than 35kDa. The mismatch between the size label and the actual protein size is obviously wrong. The authors need to explain the mismatch of the size and labeling. Also, it is better to use names instead of numbers to recognize them at a glance.

Although it is not uncommon that proteins run slightly different from their expected size in SDS-PAGE, we have repeated the experiment and run the samples in direct comparison, and in combination with a new batch of protein size marker. This marker (BlueClassic, Jena Bioscience) matches the expected protein sizes more closely and we now have replaced the blot in Suppl. Fig. 2 with the new data. In addition, we provide the Ponceau staining of the membrane as a loading control, and now directly indicate the strain genotypes in the figure.

5. When switching from pBAD to pACYC, the expression of the anchor protein changes to low. In the LITESEC system using pACYC, secretion increases in light in act3 and suppression in supp2. This is opposite to the result with the same low copies vector. Could the author explain what this interpretation is?

Indeed, the lower expression level of the anchor protein has a much less pronounced impact on the LITESEC-supp2 strain (where secretion remains strong in the dark and weak in the light) than on the LITESEC-act3 variant. We agree that this finding is relevant and now discuss it in the Discussion part (lines 423-426).

6. The pACYC184 plasmid is known to have a high copy number, but in supple figure 4, pACYC184 shows a lower expression than pBAD. However, prior to knowing this result of experiment, it was referred to each of pBAD / pACYC184 as medium-high / low copy in figure 4. It may be confusing to say the results of the experiment later, so it may be good to remove the mention or change the composition of the figures.

pACYC184 (Chang and Cohen, J Bact 1978) has been described as a low to medium-copy vector for *Yersinia* (Price et al., J Bact 1991; Kakosche et al, Sci Rep 2016). In agreement with this, pACYC184 consistently yields lower expression levels than pBAD at induction levels of 0.1-0.2% arabinose in our hands. To avoid any ambiguity, we have replaced “copy number” by “expression” throughout the text. As suggested by the reviewer, we now refer to the determination of expression levels (Suppl. Fig. 4) prior to introducing Fig. 4.

7. It is good to highlight LITESEC-act3 in Figure 5B, but it is not good to indicate the size of the circle from act2 or the position of * / ** / *** differently. Also, the L / D ratio between 5B and 5C is about 5B, which is closer to 5C.

We have adjusted the composition of Fig. 5 to ensure a clear assignment of labels, and now represent the data points more consistently, as requested.

8. In the single image of the figure, there is light/dark on the top (ex. Figure 3) and on the bottom (ex. suppl 4). When the images are unified in one position, it would be easier to understand the figure.

We have now unified the position of the light / dark labels at the top of the figures for gels and at the bottom for graphs.

9. In figure 7, both act and supp are shown to be reversible as the authors describe the advantages of the LITESEC system. However, it would be better to add the image using fluorescence.

Fig. 7 shows that effector secretion can efficiently be switched on and off in both versions of the LITESEC system. As requested by the reviewer, we now quantify export levels using the luminescence of the NanoLuc luciferase exported as a T3SS substrate, normalized by the optical density of the culture at the time of secretion, in Fig. 7B.

10. When authors see the same system in Figure 8A and the change in the dark and light states, authors should observe the same cells in the same location. As the author noted that there is a basal effect, it is not clear if the imaging position is different and the change is large enough because of the basal effect.

To visualize protein translocation into eukaryotic cells, we applied the CCF2 beta-lactamase reporter system, which required that the staining was performed after the protein translocation. Hence, we could not compare the same cells in this experiments. However, this experiment was performed on several large fields of view per strain and condition (at least 8 fields of view for controls; at least 25 fields of view for the LITESEC strains), and in independent experiments, where all controls were run in parallel. The fields of view were chosen in the DIC channel, preventing any bias for translocation efficiency (which was subsequently determined in the fluorescence channel) and all imaged fields of view are included in the analysis. Translocation into at least 671 cells per strain and condition (at least 2226 cells per strain and condition for the LITESEC strains) was quantified both by classification and automated fluorescence ratio determination, with highly comparable results. We now highlight these considerations more clearly in the Results part (lines 335-337) and the Supplementary Methods.

Reviewer #2 (Remarks to the Author):

Lindner et al describe the development of a beautiful method to induce or repress the secretion activity of bacterial type III secretion systems by optogenetic control. While I admire the idea and its development, I miss the presentation of a true proof of concept, a used case. The authors discuss the potential applications of the system in detail, in particular in the supplement, but do not go the one step further, to actually proof that the system would work in a "real" case. Clearly, optogenetic control has its limitations for use in deeper tissues and so the authors mainly suggest its use for cultured cells. I am afraid, however, that non-bacteriologist researchers and technicians will not enjoy working with alive bacteria in their cell culture systems, not to speak of clinical applications. Importantly, the authors do use a bacterial strain that has a much reduced pathogenicity due to the lack of type III-secreted effectors and is hampered in replication because of an *asd* mutation. But still, these bacteria express plenty of MAMPs that will influence signaling pathways inside the targeted eukaryotic cells.

Besides this general critique, the paper is very well written, in its logical build-up as well as in its technical detail. It features very clear figures and figure legends, sufficiently detailed methods (where provided, see below) as well as reasonable statistics where appropriate.

I also feel that this technique has substantial potential as a tool to investigate type III secretion system function - but the authors do not make use of it in the work described to this point.

We thank the reviewer for the positive and constructive feedback.

As requested, we have designed and performed an exciting application proof – the controlled induction of apoptosis in cultured eukaryotic host cells by the light-dependent translocation of the pro-apoptotic protein truncated human BH3 interacting-domain death agonist (tBID) as a T3SS substrate (YopE₁₋₁₃₈-tBID, new Fig. 8C-E).

The impact of bacteria on host cell signaling, e.g. by recognition of MAMPs, is a very important point. As mentioned by the reviewer, the strain used in this study has been depleted of the main virulence effectors and is a cell wall synthesis auxotroph, which cannot replicate in the absence of externally added diaminopimelic acid. Accordingly, bacteria in which secretion is not activated have little visual effect on the host cells, even after long incubation, which we now show in a new supplementary figure (Suppl. Fig. 10). In addition, we now discuss these considerations in more detail (lines 488-497 in the manuscript version with highlighted changes).

Minor points:

Fig. 1 shows a very much outdated cartoon of the T3SS with respect to the architecture of the export apparatus and of the cytoplasmic components. It would be nice if the authors update the cartoon appropriately.

We have updated the cartoon, based on recent structural analyses of the export apparatus (Kuhlen et al. 2018), latest information on the inner rod (Torres-Vargas et al, 2019), as well as the structure of the T3SS-bound cytosolic components (Tachiyama et al 2019).

Fig. 5C: The labeling of the Western blot image is not very clear. I assume that the labeling of Fig. 5A also applies to Fig. 5C but a repetition of the labeling or some other structuring element might makes this more clear.

The reviewer is right. As also requested by reviewer 1, we have changed the labeling, and made it more consistent across all figures.

Line 228: "inlight" should be "in light", I assume.

Corrected, thanks.

Suppl. Methods: I miss the description of the SDS PAGE and Western blot techniques used. Antibodies and their concentrations are not provided, nor the specific detection and quantification methods used. In case the authors use ECL-based immunodetection, I I question the reliability of the quantification provided in the respective figures.

We thank the reviewer for the notice. We have now added the description of the mentioned techniques, as well as the techniques used for the additional experiments in the revised version. The ECL-based immunodetection was performed in a chemiluminescence imager. We took care not to saturate any part of any image, which should provide a relatively linear quantification of the signal. Nevertheless, as requested by the reviewer, we have removed the numerical estimates of the calculated anchor/bait ratio throughout the text and now focus on the quantification-independent result that there is a clear sweet spot for the anchor/bait ratio, which we consider a helpful fact for further applications of optogenetic interaction switches in prokaryotes.

REVIEWERS' COMMENTS:

Reviewer #1 (Remarks to the Author):

All my requests have been revised.
Thank you for your effort.

Reviewer #2 (Remarks to the Author):

The authors have substantially improved the manuscript by clarifying a number of points and by showcasing an application proof. They also responded satisfactory to all of the reviewers suggestions. I support publication of this work in this revised form.